# The Heterogeneous Effects of Multilevel Centers on Farmland Transfer: Evidence from Tai'an Prefecture, China

**Meng Yang** [1,2,3], **Ting Sun** [4] **and Tao Liu** [1,2,3,*]

1   College of Urban and Environmental Sciences, Peking University, Beijing 100871, China; pkuyangmeng@pku.edu.cn
2   Center for Urban Future Research, Peking University, Beijing 100871, China
3   Key Laboratory of Territorial Spatial Planning and Development-Protection, Ministry of Natural Resources, Beijing 100871, China
4   China Academy of Urban Planning and Design (Shenzhen), Shenzhen 518034, China; st@szcaupd.com
*   Correspondence: liutao@pku.edu.cn

**Abstract:** Land transfer is an important means to achieve agricultural scale production and improve land use efficiency, as well as an effective way to solve food security issues. Discussing the mechanism of how the multilevel urban centers affect rural farmland transfer can help understand the spatial heterogeneity characteristics of farmland transfer. It is helpful to provide more policy suggestions from the perspective of urban-rural spatial relations and achieve the goal of agricultural and rural modernization. Taking Tai'an prefecture as an example, this study examines the impact of multilevel urban centers on farmland transfer by mediating effect model. The results show that: (1) Distances to urban centers are negatively associated with rural farmland transfer rates, with lower rates farther from urban centers. There are two mechanisms about how the distances to urban centers influence farmland transfer: the first is that the farther a village is from urban centers, the lower the value of its farmland, which leads to lower benefits to those who transferring farmland; the second is that lower opportunity costs of agricultural labor in the villages which farther from cities increase household reliance on farmland, reducing the rates of transferring farmland out. (2) Multilevel centers differentially influence transfers. The higher-level prefectural centers affect farmland transfer through planting structure, while the lower-level county centers affect farmland transfer through off-farm employment. Additionally, the influence of county centers is less stable due to road accessibility. (3) It is critical to additional policy support to both towns and remote villages. Particular focus should be placed on increasing the non-agricultural industries and expanding the agricultural markets of towns. It is also important to enhance infrastructure development to encourage farmland transfer in remote villages.

**Keywords:** farmland transfer; distance to urban centers; off-farm employment; planting structure; village-level data

## 1. Introduction

How to efficiently utilize farmland resources for agricultural production and solve the food security problem has always been a hot topic in rural land research globally [1–3]. Large-scale farming is an effective way to efficiently utilize farmland resources, which is conducive to advancing new technologies [4], improving mechanization levels [5], reducing farmland abandonment [6], and enhancing agricultural production efficiency [7]. Therefore, increasing the level of large-scale farming through farmland transfer has always been an important topic discussed by scholars from various countries [7–9].

China has a large population but limited farmland resources, so it is a critical issue to use limited farmland resources to ensure food security, which has also attracted attention from all over the world for a long time [10,11]. China has adopted the collective ownership system of rural farmland. After the implementation of the Household Responsibility

System in 1978, the farmland was divided and allocated to corresponding farmers in China, which led to the fragmentation of land ownership [12]. In recent years, the rural revitalization policy has proposed "modernization of agriculture and rural area". There has been a concerted effort to enhance the level of mechanized agricultural production via farmland transfer. This not only fosters large-scale farming but also addresses the quandary of fragmented land contract rights, consequently elevating agricultural productivity [13]. In China, farmland transfer refers to the transfer of farmland management rights from farmers to other farmers or economic organizations. It is an important means of large-scale farming, which is conducive to improving mechanization levels, improving farmland use efficiency, and ensuring food security [14]. Studying land transfer issues has important practical and policy significance.

The distance between urban centers and villages is an important factor in the study of farmland transfer, which is an inherent locational characteristic of villages. How to overcome the negative impact of unfavorable locations on villages is a topic of both theoretical and practical significance. Most studies argue that farmland transfer is more difficult to occur in remote villages because farmers in peri-urban areas can receive more information and opportunities for off-farm employment. Consequently, driven by the differential in benefits between non-agricultural and agricultural endeavors, these farmers choose off-farm employment, relinquish their contracted lands, and participate in farmland transfer [15,16]. On the contrary, the farmers in remote villages are more dependent on agricultural production, and their enthusiasm for participating in farmland transfer is less. However, some studies have also found that with the urban sprawl development, the conflict between construction land and agricultural land intensifies, and the value of farmland decreases, resulting in a lower farmland transfer rate in peri-urban villages [17]. Discussing the mechanism of how the distance to urban centers affects rural farmland transfer can help resolve the above research contradictions, and provide more policy suggestions for raising the farmland transfer rate in remote villages and achieving large-scale farming.

Despite the considerable attention devoted to the dynamics of farmland transfer in the extant literature, and the exploration of the influence and mechanisms of distance to urban centers on these dynamics, several issues remain unresolved. Firstly, there is a lack of consensus regarding the impact of distance to urban centers on farmland transfer. The studies anchored in the theory of location argue that the closer to the city, the lower the transportation cost, which is conducive to cultivating crops with high prices but not resistant to transportation and storage. Consequently, the agricultural economic returns from such lands are heightened, increasing farmland transfer [18]. On the other hand, studies based on decision-making mechanisms found that theoretically, the closer to the city, the more transaction information farmers can obtain, reducing the transaction cost of farmland transfer and increasing the farmland transfer rate. However, empirical evidence does not robustly present a significant influence of distance to urban centers on farmers' farmland transfer decisions [19]. These different conclusions indicate that more evidence is needed for the impact of distance to urban centers on farmland transfer.

Secondly, there is little information provided about the differences in the impact of multilevel urban centers on farmland transfer. There is heterogeneity in the off-farm employment and agricultural markets of cities at different levels. This heterogeneity inherently influences the radiative effect that urban centers have on farmland transfer. Yet, there is a paucity of research exploring the different impacts of multilevel urban centers on rural farmland transfer. Exploring the heterogeneity is helpful in understanding the influence of different levels of accessibility on rural farmland transfer.

Lastly, the mechanism of how distance to urban centers affects farmland transfer needs more empirical analysis. Although recent studies related to farmland transfer have noticed that distance to urban centers will affect rural farmland transfer and proposed theoretical explanations, they lack empirical testing. This question needs to be solved.

Furthermore, the village level has been conspicuously overlooked in studies of farmland transfer [20]. While a multitude of studies has delved into the determinant factors and

mechanisms of farmland transfer decisions at the household or individual level [21–23], and many have elucidated the macroscopic features of farmland transfer at the regional level in China [24], there is also a subset of research focusing on parcel-level factors [25]. However, the study of village-level farmland transfer characteristics and mechanisms is meaningful. Firstly, farmlands within the same village tend to share similar natural features [20]. Secondly, according to the research on farmers' behavior based on the herd effect theory, it is similar in the way agricultural production and employment choices among farmers within the same village [26]. Thirdly, the variation in farmland values within one village is minimal within the same period, resulting in relatively stable prices of farmland [27].

This study aims to examine the impact and mechanisms of distance to urban centers on village-level farmland transfer. The results of this study can help discern the relationship between the evolution of farmland transfer and urban development within the context of urban-rural integration in China. Such insights would subsequently provide more policy recommendations to promote rural revitalization and foster the modernization of agricultural production.

## 2. Framework and Hypotheses

A voluminous literature has offered valuable insights into the mechanisms of farmland transfer from a cost-benefit perspective and arrived at a fairly consistent conclusion: the rates of farmland transfer increase when the costs of farmland transfer for both transferring in and out decrease and the benefits of farmland transfer increase [28–31]. The costs and benefits of farmland transfer for both sides are influenced by the distance to urban centers. For instance, the classic locational theory suggests that the closer to the city, the lower the transportation costs of agricultural production, the more the economic benefits of agricultural land, and thus the more transferees to transfer in [32]; while the classic dual sector model and the Todaro model suggest that the closer to the city, the more off-farm employment opportunities rural households can obtain, the higher the opportunity costs of agricultural labors, rural households will reduce or abandon agricultural production, and obtain more economic benefits from transferring land out [33]. Figure 1 illustrates these two possible mechanisms of distance to urban centers affecting village land transfer: (a) Agricultural Production Hypothesis; and (b) Off-farm Employment Hypothesis.

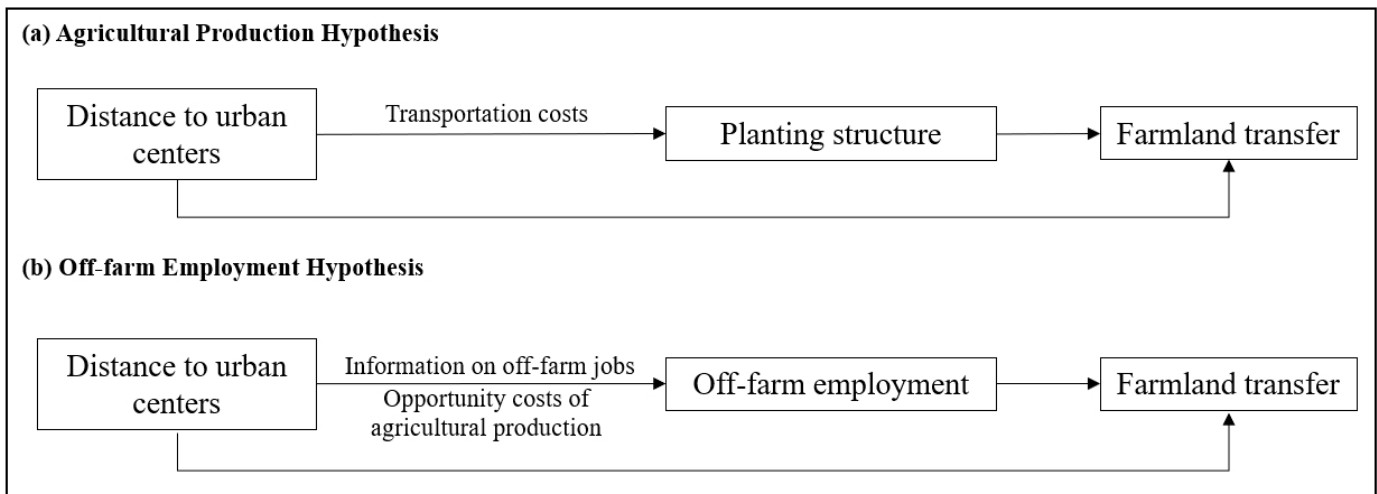

**Figure 1.** Analysis framework.

While there exists heterogeneity in the impact mechanisms of distance to urban centers on both transferring in and out, their interrelated interests remain consistent. As the distance to urban centers increases, the costs associated with transferring farmland out increase [34], and the benefits of transferring farmland decrease [35]. Consequently, the

distance to urban centers has a negative impact on both decisions of farmland transferring in and out, resulting in a lower rate of rural farmland transfer as the distance increases.

Therefore, this study proposes the first hypothesis: The rate of farmland transfer in the village decreases as the distance to urban centers increases, with the village located farther from urban centers exhibiting a lower farmland transfer rate.

Location is the basic geographic characteristic of land, which affects farmland transfer by influencing the benefits of agricultural production of those who transfer farmland [36–38]. The classic agricultural location theory and rent theory argue that the city is the market for agricultural products. The closer to the urban centers, the stronger the market accessibility, which means the lower the transportation costs of agricultural products to the markets. Therefore, crops that are not resistant to transportation and storage but have more economic value tend to gravitate toward the villages near markets [17,18,36]. Under the influence of the homogenization of planting structures and high farmland rents, the degree of intensification of farmland use increases, and the negative impact of high land rents is offset by reducing the marginal cost of agricultural production input [39]. As a result, in the villages surrounding urban centers, the agglomeration of agricultural production of crops with high economic value is facilitated through farmland transfer.

Based on this, this study proposes the second hypothesis: The economic value of farmland decreases with distance to urban centers increasing. As villages are located farther from urban centers, the economic benefits from transferring in are less, leading to a reduced rate of farmland transfer.

In recent years, the studies of rural farmland transfer have also begun to pay more attention to the impact of cities on rural households' decision of farmland transfer [40,41]. The off-farm employment opportunities in the city have a spatial spillover effect [41], and rural households in the villages closer to urban centers can obtain more off-farm employment information and opportunities. Due to the higher benefits of off-farm employment, the opportunity costs of agricultural labor increase, leading rural households to reduce the agricultural input, thereby resulting in farmland abandonment. Abandoning farmland will not bring more benefits to rural households and even increase the risk of being punished [42]. With land rents being comparatively higher near urban areas, the economic incentives drive the households to positively transfer their lands out to these villages. In contrast, households in villages located farther from urban centers have limited access to information about off-farm employment opportunities. The commuting costs of off-farm jobs escalate, reducing the opportunity costs associated with agricultural production [43]. As a result, these households rely more on the agricultural output of their lands and are less likely to transfer farmland out.

Drawing upon the analysis, this study proposes the third hypothesis: As the village is closer to urban centers, the opportunity costs of agricultural labor are higher, leading to an increased rate of farmland transfer. Conversely, as the distance to urban centers increases, rural households rely more on agricultural production, resulting in a decreased rate of farmland transfer.

The urban markets, both for agricultural products and off-farm employment, exert influences on farmland transfer. There exists heterogeneity in the scale of off-farm employment and agricultural markets across different levels of cities [44]. For example, in low-level cities, the proportion of the non-agricultural industry is lower, so it is difficult to promote off-farm employment in these villages and has little influence on rural farmland transfer; while in higher-level cities, the proportion of industrial and service industries is higher and the scale of agricultural market is larger, which not only has a positive effect on rural households' off-farm employment, but also affects the planting structure of suburban villages. Therefore, the influence of the distances to different levels of cities on farmland transfer may be different.

In light of the preceding analysis, this study proposes the fourth hypothesis: There exists a heterogeneous influence of distances to multilevel cities. Specifically, the impact

of distance to high-level cities on farmland transfer is stronger compared to that of low-level cities.

Beyond distance to urban centers, other determinant factors such as topography, resource endowment, economic development, population structures, and policy also have impacts that cannot be ignored and should be considered in the empirical analysis. Existing literature has discussed the impact of topography, land quality, and cultivated land area on farmland transfer [31,45,46]. It has been proved that when natural conditions are more favorable and farmland resources are abundant, the rate of farmland transfer increases. Such conditions can diminish the costs of large-scale agricultural production, yielding greater outputs for the same inputs [47]. Some studies have also discussed the impact of economic development on land transfer and found that regional economics or household income has a positive effect on farmland transfer [37,48]. As the income of a rural household increases, the possibility of off-farm employment increases, leading to a greater inclination towards farmland transfer [7], and it is the same in region-level studies [49]. When there is more male and labor population, the comprehensive labor capacity of a household is enhanced, increasing their dependency on land and consequently lowering the rate of farmland transfer [50]. China's land system is different from Western countries. In Western countries, farmland is privately owned and ownership can be determined by individuals. While in China, farmland is collectively owned [51]. It means that the policy may have a stronger impact on land use in China. When villages benefit from more favorable policies, incentives are generated for the agricultural producers, propelling them to expand the scale of agricultural production and thereby amplifying the rate of farmland transfer [52,53]. As such, when examining the mechanisms through which distance to urban centers impacts farmland transfer, these factors also need to be reflected in empirical analysis.

## 3. Data and Methods

### 3.1. Study Area

This study takes Tai'an Prefecture as the case (Figure 2). Tai'an is located in the middle of Shandong Province. The terrain is mainly plains and hills, with a few mountains in the north and east. Tai'an is surrounded by Jinan Prefecture, Jining Prefecture, Laiwu Prefecture, and Liaocheng Prefecture. It encompasses over 3000 villages spread across six county-level regions: Taishan District, Daiyue District, Xintai City, Feicheng City, Ningyang County, and Dongping County. In 2020, the urbanization rate of Tai'an's population is approximately 64%, representing a 14-percentage point increase over the past decade. This denotes a significant transformation in the urban-rural relationship. Nonetheless, the agricultural industry plays an important role in the economic system. According to the statistical data of Tai'an in 2020, the total agricultural output value of Tai'an increased by 2.5% compared with last year, the grain yield per unit area is 0.69 tons/hectare, ranking second in the province.

The suitable natural geographic conditions and diversified topographic features indicate that Tai'an has the potential for large-scale agricultural production and may have more cases for farmland transfer on the one hand, and on the other hand, Tai'an is a place with high-speed urbanization and a significant transformation in the urban-rural relationship but still keeps a high-level agricultural production, which may be relative to farmland transfer. These characteristics make Tai'an a proper case for exploring the impact of distance to urban centers on farmland transfer.

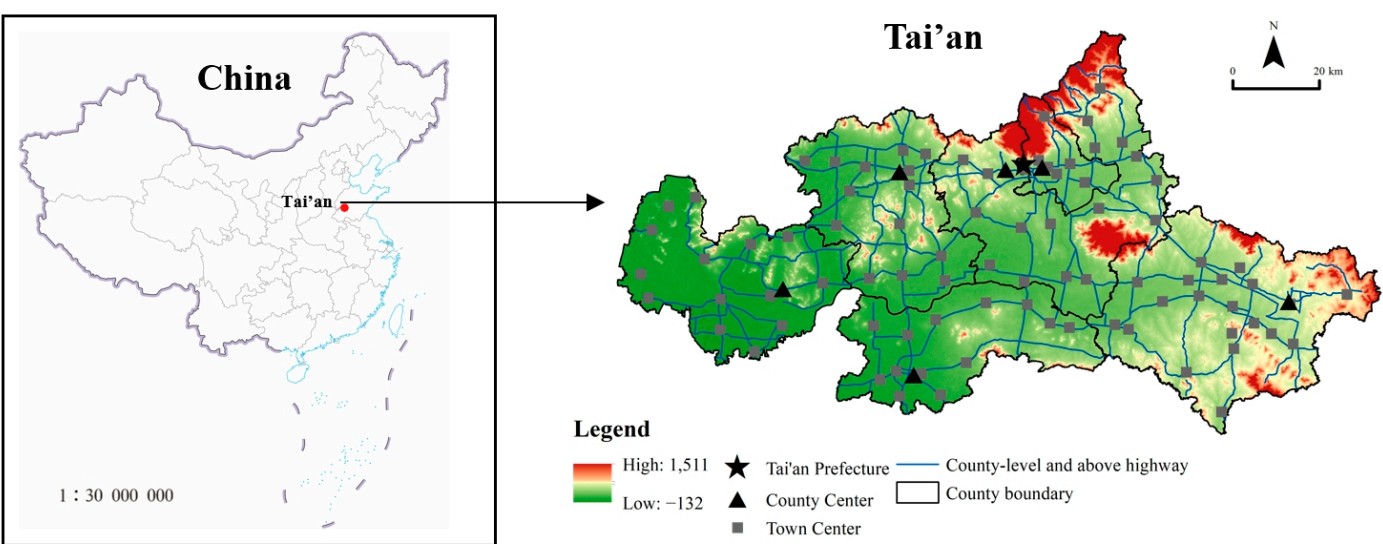

**Figure 2.** Physical geography and location of Tai'an.

*3.2. Data and Model Specification*

This study uses several sets of data, including farmland transfer data, geographical data, geophysical characteristics of farmland, economic data, demographic data, and policy. Considering that the period of the farmland transfer data is 2015, most of the datasets are representative of this year. However, due to constraints in data sources, some datasets are representative of 2019. The land transfer data, economic data, and population structure data come from the 2015 village report of Tai'an Prefecture, the geophysical characteristics of farmland are derived from the third national land survey in China, the geographical data are obtained from the Digital Elevation Model (DEM) on the official website of the Geospatial Data Cloud, which shows the land use in 2015, and the policy refers to the historical and cultural towns and villages of Shandong Province and the first batch of rural revitalization demonstration villages announced by Shandong Province before 2019.

In this study, the direct effect of distance to urban centers on farmland transfer and the mediating effect of planting structure and off-farm employment are tested by the Causal Steps Approach [54–56]:

$$Y = \alpha_0 + \alpha_1 X + \delta Z + \varepsilon \tag{1}$$

$$M_1 = \beta_0 + \beta_1 X + \delta Z + \omega \tag{2}$$

$$M_2 = \beta'_0 + \beta'_1 X + \delta' Z + \omega' \tag{3}$$

$$Y = \gamma_0 + \gamma_1 X + \gamma_2 M_1 + \gamma_3 M_2 + \delta Z + \sigma \tag{4}$$

In Formulas (1)–(4), $Y$ represents the farmland transfer rate, $X$ represents the distance to urban centers, which initially took into account the Euclidean distance, but after modifying the model, the road distance was taken into account, $M_1$ and $M_2$, respectively represent the planting structure and off-farm employment, $Z$ represents a series of control variables, $\alpha$, $\beta$, $\beta'$, $\gamma$ and $\delta$ are the estimated coefficients, $\varepsilon$, $\omega$, $\omega'$ and $\sigma$ are the random error terms.

The first step is to judge whether the total effect ($\alpha_1$) is significant. The second and third step is to test whether the direct effect and mediating effect ($\gamma_1$, $\beta_1\gamma_2$ and $\beta'_1\gamma_3$) are significant. Because there are two mediators in this study, it is necessary to repeat Formula (1) twice using different control variables. For more accurate estimates, the percentile and bias-corrected Bootstrap test is used to test the robustness of the mediating effects [56].

*3.3. Variables*

Table 1 lists the datasets and summary statistics of variables. The dependent variable in this study is the farmland transfer rate. This rate is derived by dividing the number of rural households participating in farmland transfer by the total number of rural households in the village. It can better reflect the level of activity of farmers' participation in farmland transfer. A higher farmland transfer rate indicates more active participation of rural households in farmland transfer, meaning a higher level of farmland transfer within the village. The farmland transfer rate of some villages is 1, which means all the farmland of this village is transferred out.

**Table 1.** Summary statistics of variables.

| Type | Variable | Description | Min | Max | Mean | Standard Deviation |
|---|---|---|---|---|---|---|
| Dependent variable | Farmland transfer rate | The rate of households who participate in farmland transfer to the total households | 0 | 1 | 0.236 | 0.269 |
| Independent variables | Distance_P | Distance to the nearest prefecture-level center (km) | 2.15 | 64.35 | 36.97 | 13.79 |
| | Distance_C | Distance to the nearest county-level center (km) | 0.52 | 44.01 | 18.173 | 9.177 |
| | Distance_T | Distance to the nearest town-level center (km) | 0.01 | 11.70 | 4.054 | 1.982 |
| Mediating Variables | Off-farm employment | The proportion of off-farm employment to the total workers | 0 | 1 | 0.699 | 0.190 |
| | Planting structure | The proportion of vegetable planting area to the total planting area | 0 | 1 | 0.189 | 0.195 |
| Control variables: | | | | | | |
| Natural condition | Terrain index | The index according to the terrain index formula [57] | 0.29 | 1.36 | 0.574 | 0.169 |
| | Quality | The proportion of the area of high-quality farmland to the total farmland area | 0 | 1 | 0.725 | 0.370 |
| | Integrity | Average area of farmland plot (acre) | 0.90 | 495.10 | 40.213 | 29.802 |
| | Farmland | The proportion of farmland area to the total area | 0.01 | 0.99 | 0.777 | 0.128 |
| Transportation location | Distance_H | Distance to the nearest county-level and above highways (km) | 0 | 16.64 | 2.826 | 2.778 |
| Economic factor | Electricity | Per capita electricity consumption (10,000 kWh/person) | 0.02 | 0.05 | 0.023 | 0.039 |
| Population structure | Male | The proportion of males to the total population | 0.33 | 0.74 | 0.515 | 0.033 |
| | Labor | The proportion of the labor to the total population | 0.15 | 1 | 0.619 | 0.106 |
| Policy | Policy support | Whether it is a historical and cultural village or a demonstration village for rural revitalization | 0 | 1 | - | - |

The distances from villages to the nearest prefectural center, county center, and town center are three independent variables in this study. This study categorizes urban levels based on China's administrative division levels. Within the Chinese administrative hierarchy, prefectures, counties, and townships are classified as the second, third, and fourth levels. The level of the administrative division correlates with its socio-economic status. Typically, a higher-level urban center implies a more advanced economic level, a larger

consumer market, and a more expansive non-agricultural employment market [41]. Given that in China, the location of governments largely coincides with market centers [20], thus the urban centers are represented by the governments of prefectures, counties, and towns in this study. This serves as an indicator of the spatial barriers that cities influence the rural farmland transfer. The higher the level of the urban centers, the larger the scale of off-farm employment and agricultural markets. Consequently, there is more negative impact of the distance to the high-level urban centers on rural farmland transfer. Some villages are in the towns but have farmland, and the distances to the town centers are very short.

The mediating variables in this study are the off-farm employment rate and the vegetable planting rate. As the distance to the urban centers decreases, the opportunity costs of agricultural labor rise, leading to a higher off-farm employment rate [58,59]. A higher off-farm employment rate leads to an increased farmland transfer rate. The proportion of off-farm employment in some villages is 1, which means that all the workers have an off-farm job, and the other family members, such as the old people, may do the agricultural work. If there is no family member to do the agricultural work, the probability of households transferring out their farmland will increase. Vegetables are characterized by high economic prices and are not resilient to transport and storage, thus it is suitable for making the vegetable planting rate the variable of planting structure [60]. As the distance to the urban centers decreases, the vegetable planting rate escalates, indicating a higher economic value of the farmland and an increased farmland transfer rate.

The control variables include the following types: natural conditions, transportation locations, economic factors, population structure, and policy. Natural conditions have a direct impact on agricultural production, determining the cost and benefit of agricultural production, thus influencing farmland transfer decisions. The terrain index, the average area of farmland plots, and the proportion of high-quality farmland are used to measure the natural conditions of farmland. In addition, the proportion of farmland area to the total area is used to characterize the abundance of farmland resources. Transportation means the external traffic conditions of the village. The better the transportation, the more convenient the village is to the urban centers, leading to a higher farmland transfer rate [44]. However, some studies have found that roads will cause farmland fragmentation, and the associated facilities will affect the agricultural production conditions of nearby villages, thus having a negative impact on the farmland transfer rate [61]. The distance to the nearest county-level and above highways is used to reflect the traffic situation according to the previous studies [20,62,63]. Economic characterizes the economic level of the village. According to previous studies and the data source, this study uses per capita electricity consumption to reflect the economic level of the village [64]. Population structure affects the agricultural production activities of the village, thus having an impact on the farmland transfer. This paper uses the male rate and labor force rate to characterize gender [44] and age structure [65]. It is expected that the higher the male rate and labor force rate, the stronger the production capacity of farmers with multiple occupations, and the less inclined to participate in land transfer. Policy type is whether the village is a historical and cultural famous village or a rural revitalization demonstration village before 2019. These villages will receive better policy support and more economic subsidies from the local governments [20], which promotes the development of large-scale farming and positively influences households' participation in farmland transfer. This is a binary variable that 1 means the village is a historical and cultural village or a demonstration village for rural revitalization and 0 means it is not.

## 4. Results

### 4.1. The Farmland Transfer and Distance to the Urban Centers

In Tai'an Prefecture, farmland transfer is relatively active. After excluding the villages with missing data and anomalies (where the number of households participating in the land circulation exceeds the total number of households), there are 3322 valid samples. Out of these, farmland transfer occurred in 2390 villages, representing approximately 71.94%

of the samples. The average farmland transfer rate is about 23.15%, with the highest rate reaching 100%. The spatial distribution is shown in Figure 3. Notably, villages in the central and southern plains exhibit higher rates of land circulation participation. Most county centers are located in areas with a high concentration of farmer participation. Conversely, a greater number of villages in the western and eastern regions have a 0% participation rate, and these villages tend to be far from the county centers.

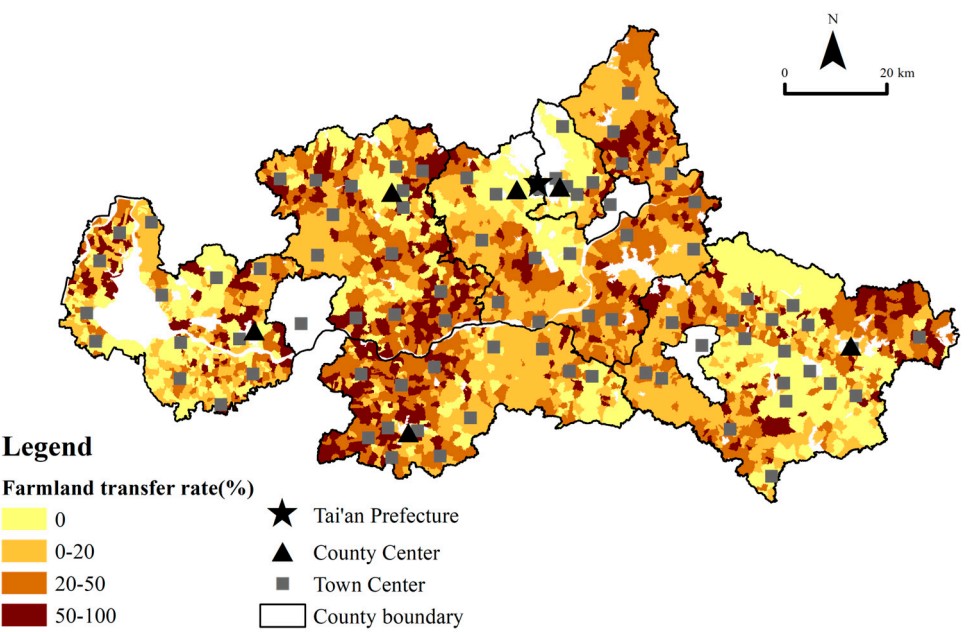

**Figure 3.** The spatial distribution of farmland transfer rates.

Owing to the limitations of traditional scatterplots in highlighting the relationship between dependent and independent variables when there are large numbers of samples, this study employs binscatter to delineate the relationship between farmland transfer rates and the distance to multilevel centers [66]. Through showing the correlation between the distance to the nearest prefectural center, county center, and town center and the farmland transfer rate by binscatter, it is found that as the distance from a village to the nearest prefectural and county center increases, there is a significant decline in the farmland transfer rate (Figure 4a,b). However, as the distance from a village to the nearest town center increases, the farmland transfer rate does not evidently (Figure 4c).

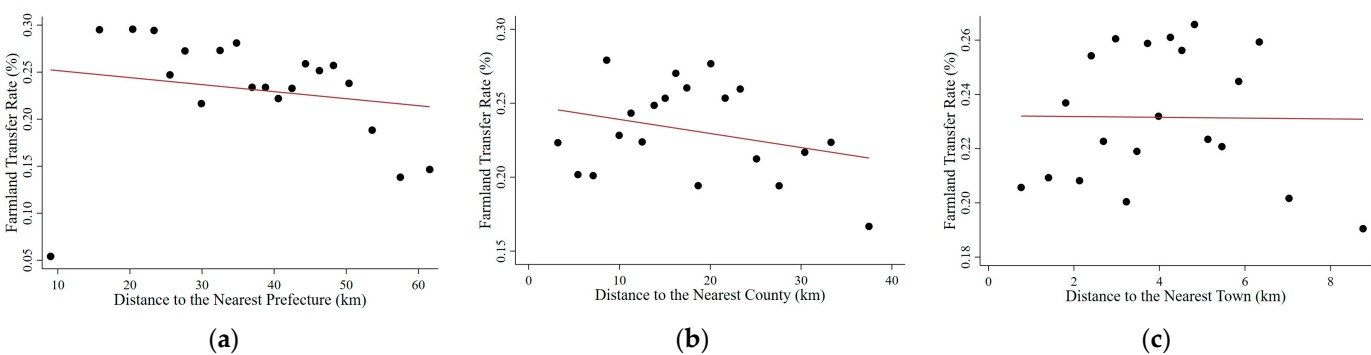

**Figure 4.** The correlation between the distance to the nearest urban center and farmland transfer rate: (**a**) Prefectural center; (**b**) County center; (**c**) Town center.

This is consistent with the existing research results, which indicate that the farther away the village is from the city, especially the prefectural center and county center, the lower the probability of farmland transfer [15]. Moreover, the absolute value of the slope

of the fitted line representing the relationship between the distance to the nearest town center and the farmland transfer rate is markedly smaller than that for the distance to the nearest prefectural and county center. This indicates that lower-level urban centers do not exhibit as pronounced an influence as higher-level urban centers. Such a result can explain why extant studies suggest that the distance to town centers has no significant effect on farmers' willingness and decision-making concerning farmland transfer [67]. The presence of high-level prefectures and counties seems to diminish the influence of low-level towns, leading to a nearly horizontal line of the correlation between the distance to the nearest town center and the farmland transfer rate. Nevertheless, it is necessary to examine the effects of three urban centers on farmland transfer together because the fitted line canot refuse the hypothesis that there are mediating effects between town centers and rural farmland transfer. Thus, it is necessary to prove the hypotheses by the mediating effect model.

### 4.2. Mechanism Analysis

In Section 4.1, through spatial distribution analysis and linear regression, it is discerned that the distance from a village to an urban center, especially the prefectural and county center, might influence the farmland transfer rate. However, this preliminary conclusion needs further rigorous model testing. Consequently, Section 4.2 empirically tests whether the fitting results are valid and calculates the proportion of the mediating effect to the total effect.

Based on the results from Models 1, 2, and 5, it is observed that compared to prefectural and town centers, county centers exert a more pronounced impact on the off-farm employment in villages, subsequently influencing the rural farmland transfer rates. In Model 1, town centers do not exhibit a significant total effect on rural farmland transfer rates at a 90% confidence level. While prefectural centers demonstrate a significant total effect in Model 1, the relationship between prefectural centers and off-farm employment does not pass the significant test at a 90% confidence level in Model 2. Only county centers display the anticipated impact consistently across Models 1, 2, and 5, highlighting the unique role of county centers as urban centers.

This suggests that the higher urban hierarchy does not necessarily translate into a stronger capacity to stimulate the rural population in suburbs to engage in off-farm employment. This finding confirms previous studies suggesting that prefectural centers do not always play a role in promoting off-farm employment in suburbs [68]. Compared to prefectural centers, living in county centers needs a lower level of living costs, and there is a greater emphasis on off-farm employment opportunities that do not necessitate advanced skills, such as those found in the mining and manufacturing sectors in counties [15,20]. Additionally, land rents in villages around counties are lower than those around prefectural centers [15], further facilitating the transition to localized off-farm development.

Models 3, 4, and 5, which explore the mediating path of planting structure, reveal that both county and prefectural centers influence rural farmland transfer through planting structure. However, based on both statistical significance and correlation coefficients, the impact of the prefectural centers is found to be more potent and stable. This indicates two points: firstly, the classic agricultural location theory remains applicable to contemporary rural agricultural studies, and secondly, with the advancements in transportation infrastructure, urban centers still have a spatial influence on rural agriculture. This spatial impact is even more pronounced for urban centers of a higher level. Thus, Prefectural centers possess a larger agricultural product consumption market. Moreover, compared to county centers, these prefectural centers have a higher population density, superior transportation facilities, and an ample consumer base [14]. Such factors favor the development of sight-seeing agriculture and leisure agriculture in the surrounding villages, and these types of agriculture require more farmland to create agricultural landscapes and offer ample spaces for visitors, which is highly correlated with farmland transfer.

The total effects of the distance to the nearest town center do not show statistical significance at a 90% confidence level. Moreover, some related studies argue that town

centers do not have a significant impact on farmers' decisions of farmland transfer [68]. While villages near towns exhibit a lower proportion of off-farm employment, they possess a higher percentage of vegetable planting area, suggesting there remains a large room for improvement in the off-farm employment markets in the towns. In China, small towns often face problems such as underdeveloped non-agricultural industries and limited growth potential [69]. They fail to absorb the surplus labor force from villages, making it difficult to influence the urbanization of the rural population and the transformation of land use in surrounding villages [20,70]. When compared to prefectures and counties, towns exert a weaker influence on the farmland transfer to villages. This finding holds significant policy implications. Within the hierarchical system of county-township-village, it is necessary to harness the developmental potential of towns, propel the transformation and upgrading of their industrial structures, and enhance their radiation and driving ability in villages.

The control variables nearly show the expected effects. The villages with more suitable natural conditions for agricultural production have higher farmland transfer rates. These villages are endowed with abundant and higher-quality land resources, suitable for large-scale and mechanized farming. As a result, the farmland transfer rates are higher in these villages. However, when the fragmentation of arable land is minimal, the farmland transfer rate decreases. This is because villages with less fragmented farmland do not necessarily require transfer to consolidate land resources, leading to a reduced transfer rate [44]. The higher the economic level of a village, complemented by better infrastructure, the more conducive it is for large-scale agricultural production, and correspondingly, the higher the rate of farmland transfer. The labor force proportion does not pass statistical significance tests at a 90% confidence level, and a higher male proportion negatively affects farmland transfer. Influenced by the traditional model of "men plowing and women weaving" [63], males tend to place a higher value on land assets and are less likely to abandon agricultural production. Policy support received by villages elevates the rate of farmland transfer, indicating that policy backing and economic subsidies positively influence the development of farmland transfer in the villages.

Table 2 proves the first three hypotheses, while the fourth hypothesis is partially validated. To measure the percentage of the mediating effect of the distance to the nearest county center more accurately, this study calculates the proportion of the mediating effect of each variable (Table 3). The results show that about Distance_P, the mediating paths of off-farm employment and planting structure account for 2.33% and 33.45% partly, but only planting structure is statistically significant at a 99% confidence level. About Distance_C, the mediating paths of off-farm employment and planting structure account for 9.28% and 22.27% partly, and the total mediating effect is 31.55%. Numerically, the mediating effect of planting structure has a higher proportion.

The Causal Steps Approach for testing the mediating effect has been criticized for its insufficient test power. To enhance the robustness, this study adopts the percentile and bias-corrected Bootstrap method with stronger test power to test the two mediating paths of the urban distance and sets the resampling times to 5000 according to previous studies [71]. Table 4 shows that the planting structure of county centers cannot pass the test because the 95% confidence intervals include 0, and the other paths pass the test.

This indicates that the mediating effect of county centers on rural farmland transfer rates through planting structure is not robust. Compared to county centers, prefectural centers represent a higher-level market center, possessing a larger agricultural product consumption market. Under the influence of prefectural centers, the path of which county centers affect rural farmland transfer rates through planting structures becomes unstable. Existing studies have also identified that in prefectural centers, there are more consumers for new types of agriculture, such as sightseeing agriculture and leisure agriculture [14,53]. Lower-level urban centers, such as county centers, are unable to provide a sufficiently large consumer base for these new types of agriculture and thus have less capacity to influence rural farmland transfer. This further indicates the intensified role of higher-level urban centers in driving the transformation of village agriculture.

**Table 2.** The results of mediating effects.

| Variable | Model 1 | Model 2 | Model 3 | Model 4 | Model 5 |
|---|---|---|---|---|---|
| Distance_P | −0.001 ** | −0.000 | −0.002 *** | −0.003 *** | −0.001 ** |
| | (−2.18) | (−0.91) | (−4.05) | (−12.10) | (−2.13) |
| Distance_C | −0.001 *** | −0.002 *** | −0.001 ** | −0.001 * | −0.001 ** |
| | (−2.60) | (−4.33) | (−2.56) | (−1.69) | (−2.32) |
| Distance_T | 0.000 | 0.004 ** | −0.000 | −0.001 | 0.000 |
| | (0.17) | (2.03) | (−0.08) | (−0.76) | (0.04) |
| Off-farm employment | | | 0.094 *** | 0.007 | 0.093 *** |
| | | | (3.62) | (0.40) | (3.60) |
| Planting structure | 0.223 *** | 0.007 | | | 0.223 *** |
| | (9.15) | (0.40) | | | (9.14) |
| Terrain index | −0.246 *** | 0.282 *** | −0.272 *** | 0.002 | −0.272 *** |
| | (−7.29) | (12.38) | (−7.79) | (0.08) | (−7.90) |
| Integrity | −0.000 *** | −0.000 *** | −0.000 | 0.001 *** | −0.000 ** |
| | (−2.68) | (−3.69) | (−0.95) | (9.56) | (−2.45) |
| Quality | 0.016 | −0.083 *** | 0.013 | −0.051 *** | 0.024 * |
| | (1.26) | (−9.53) | (0.96) | (−5.53) | (1.84) |
| Farmland | 0.209 *** | −0.212 *** | 0.209 *** | −0.084 *** | 0.228 *** |
| | (5.28) | (−7.96) | (5.20) | (−2.97) | (5.73) |
| Ditance_H | 0.013 *** | 0.003 * | 0.015 *** | 0.012 *** | 0.012 *** |
| | (4.40) | (1.76) | (5.17) | (5.82) | (4.29) |
| Electricity | 0.441 *** | 0.187 ** | 0.426 *** | 0.012 | 0.423 *** |
| | (3.70) | (2.32) | (3.54) | (0.15) | (3.56) |
| Male | −0.415 *** | 0.620 *** | −0.443 *** | 0.131 | −0.473 *** |
| | (−2.95) | (6.54) | (−3.10) | (1.31) | (−3.35) |
| Labor | 0.008 | 0.195 *** | −0.007 | 0.015 | −0.011 |
| | (0.18) | (6.76) | (−0.16) | (0.50) | (−0.25) |
| Policy | 0.142 *** | 0.017 | 0.145 *** | 0.023 | 0.140 *** |
| | (3.90) | (0.68) | (3.95) | (0.87) | (3.86) |
| Constant | 0.409 *** | 0.351 *** | 0.438 *** | 0.278 *** | 0.376 *** |
| | (4.95) | (6.29) | (5.23) | (4.71) | (4.53) |
| N | 3322 | 3322 | 3322 | 3322 | 3322 |
| $R^2$ | 0.065 | 0.151 | 0.046 | 0.085 | 0.069 |

Note: *, **, and *** indicate significance at 0.1, 0.05, and 0.01 levels, respectively. The dependable variables of Models 1–5 are farmland transfer rate, off-farm employment rate, farmland transfer rate, vegetable planting rate, and farmland transfer rate, respectively.

**Table 3.** The proportion of mediating effects.

| | Variable | The Coefficient and Proportion of Effects |
|---|---|---|
| Distance_P | | −0.002 *** |
| | Off-farm employment (%) | 2.33 |
| Wherein: | Planting structure (%) | 33.45 *** |
| | Total mediating effects (%) | 35.78 (Only planting structure is significant) |
| Distance_C | | −0.001 *** |
| | Off-farm employment (%) | 9.28 *** |
| Wherein: | Planting structure (%) | 22.27 *** |
| | Total mediating effects (%) | 31.55 *** |
| Distance_T | | No significance |
| | Off-farm employment (%) | - |
| Wherein: | Planting structure (%) | - |
| | Total mediating effects (%) | No significance |

Note: *** indicate significance at 0.01 levels, respectively.

Based on Tables 2–4, a mediation effect diagram is illustrated in Figure 5, revealing differences in coefficients across the two mediating paths which pass the tests of the Causal Steps Approach and Bootstrap Method. The first path is that prefectural centers affect

the rural farmland transfer through planting structure, and the second path is that county centers affect the rural farmland transfer through off-farm employment.

**Table 4.** Bootstrap test.

| Mediating Variable | Result Type | Acting Path | Point | Bootstrap Std. Err. | Z | P > z | BC-Bootstrap Normal-Based [95% Conf. Interval] | |
|---|---|---|---|---|---|---|---|---|
| Off-farm employment | direct | Distance_C → Farmland transfer rate | −0.0012 | 0.00005 | −2.41 | 0.002 | −0.00221 | −0.00023 |
| | indirect | Distance_C → Off-farm employment → Farmland transfer rate | −0.0001 | 0.00005 | −2.86 | 0.004 | −0.00024 | −0.00004 |
| Planting structure | direct | Distance_C → Farmland transfer rate | −0.0012 | 0.00051 | −2.39 | 0.017 | −0.00222 | −0.00022 |
| | indirect | Distance_C → planting structure → Farmland transfer rate | −0.0001 | 0.00008 | −1.75 | 0.080 | −0.00030 | 0.00002 |
| Planting structure | direct | Distance_P → Farmland transfer rate | −0.0008 | 0.00039 | −2.05 | 0.040 | −0.00159 | −0.00004 |
| | indirect | Distance_P → planting structure → Farmland transfer rate | −0.0007 | 0.00010 | −7.13 | 0.000 | −0.00919 | −0.00052 |

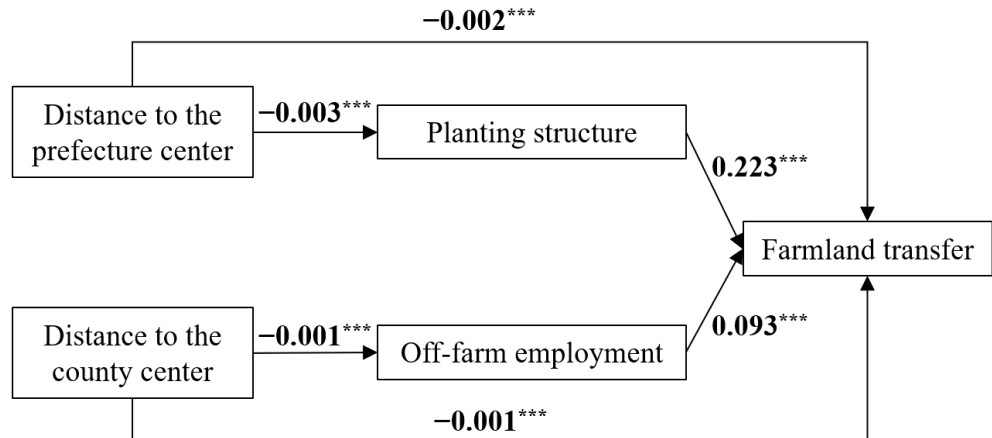

**Figure 5.** The path coefficients (*** indicate significance at 0.01 levels, respectively).

From the perspective of the coefficients of the total effects, the coefficient of prefectural centers on farmland transfer has a higher absolute value than that of county centers. This discrepancy indicates that, despite different influence paths, higher-level urban centers do indeed exert a more pronounced impact. Considering the coefficients of mediating factors, the coefficient of planting structure on farmland transfer has a higher absolute value than that of off-farm employment. Such differences suggest that in the current context in China, where the disparity in profits between non-agricultural income and agricultural income increases farmers' willingness to transfer out, maximizing the agricultural economic value of farmland and increasing the willingness of transferees to transfer farmland in become even more crucial in improving the farmland transfer rates in villages.

*4.3. Robustness Test*

The distances from the village to the prefectural, county, and town centers are Euclidean distances, which may not reflect the actual accessibility of the village to the urban centers. Therefore, this paper used the shortest traffic distances from the village to the urban centers to replace the independent variables [20]. The shortest traffic distances from the village to the urban centers are calculated using ArcGIS 10.5 software based on the data

of the road network, and then the mediating effect model is run. Table 5 shows the results of the robustness test. The results show that the influence of prefectural centers does not change much, indicating the robustness of the mediating effect. However, the total effect of county centers changes to be not significant at a 90% confidence level. This indicates that when considering the road distance to the urban centers, the interference from higher-level prefectural centers intensifies.

**Table 5.** The results of mediating effects after changing the independent variable.

| Variable | Model 6 Farmland Transfer Rate | Model 7 Off-Farm Employment Rate | Model 8 Farmland Transfer Rate | Model 9 Vegetable Planting Rate | Model 10 Farmland Transfer Rate |
|---|---|---|---|---|---|
| Distance_P | −0.001 ** (−2.53) | 0.000 (1.16) | −0.002 *** (−4.80) | −0.003 *** (−14.06) | −0.001 *** (−2.60) |
| Distance_C | −0.001 (−1.65) | −0.001 *** (−4.14) | −0.001 (−1.25) | 0.000 (0.75) | −0.001 (−1.39) |
| Distance_T | −0.001 (−0.51) | 0.003 *** (2.75) | −0.002 (−1.11) | −0.004 *** (−2.87) | −0.001 (−0.68) |
| Off-farm employment | | | Control | Control | 0.098 *** (3.76) |
| Planting structure | Control | Control | | | 0.240 *** (9.83) |
| Control variables | Yes | Yes | Yes | Yes | Yes |
| Constant | 0.447 *** (5.25) | 0.353 *** (6.18) | 0.479 *** (5.56) | 0.295 *** (4.95) | 0.414 *** (4.84) |
| N | 3322 | 3322 | 3322 | 3322 | 3322 |
| $R^2$ | 0.064 | 0.148 | 0.046 | 0.101 | 0.068 |

Note: **, and *** indicate significance at 0.05, and 0.01 levels, respectively.

After replacing the independent variables, the same Bootstrap test is conducted for the mediating effect, and the results in Table 6 show that the mediating effect still passes the test. This suggests that even when transitioning from Euclidean to traffic distance, accessibility to prefectural centers still impacts farmland transfer rates by raising the proportion of vegetable planting area.

**Table 6.** The Bootstrap test after changing the independent variable.

| Mediating Variable | Result Type | Acting Path | Point | Bootstrap Std. Err. | Z | P > z | BC-Bootstrap Normal-Based [95% Conf. Interval] | |
|---|---|---|---|---|---|---|---|---|
| Planting structure | direct | Distance_P → Farmland transfer rate | −0.00081 | 0.00040 | −2.05 | 0.040 | −0.00159 | −0.00004 |
| | indirect | Distance_P → Planting structure → Farmland transfer rate | −0.00072 | 0.00010 | −7.13 | 0.000 | −0.00092 | −0.00052 |

The results that the total effect of county centers does not pass the robustness test does not negate the meaning of prior research regarding the mediating path of county centers, as the impact of county centers on off-farm employment in the suburban villages still passes the test at a 99% confidence level. It means that the transportation of the higher-level urban centers strengthens its positive influence on rural farmland transfer, and suggests that more attention should be paid to the development of transportation infrastructure in county centers to reflect the positive role of county centers in rural farmland transfer.

## 5. Discussion

This study finds that urban centers have an impact on farmland transfer in villages, but this impact is more prominent in high-level urban centers. Proximity to the urban centers correlates with a higher eagerness among farmers to participate in farmland transfer, leading to a heightened level of farmland transfer. However, this does not mean that the studies concluding that the distance to urban centers has no impact on farmland transfer are meaningless. This study just offers a perspective to discuss the possibility that different paths take place in different places. This study discusses the related issues from the village-level perspective in Tai'an and discusses the different mediating paths of the impact of the distance to urban centers on farmland transfer in villages under varying contexts.

First, the classical agricultural location theory can explain the mechanism of urban centers on the rural farmland transfer through the planting structure. The villages in the suburbs benefit from reduced transportation costs but have high rents. To overcome the negative impact of land rents, crops that are not resistant to transportation and storage but have high economic value are homogenized and scaled up. Thus, the intensity of agricultural production is enhanced, which increases the farmland transfer rates in the villages. Conversely, the villages farther away from urban centers have more dispersed socio-economic activities, leading to diminished farming intensity and lower economic value from agriculture. Besides, there will also be new types of agriculture such as sightseeing agriculture and leisure agriculture near the cities. These new types of agriculture that are combined with the secondary and tertiary industries have more economic benefits but also need stronger urban accessibility to attract more consumers. These new types of agriculture also require capital investment and land resource agglomeration, which increases the farmland transfer rates.

Second, recent studies have also found that the city affects rural land resource reallocation by influencing the sectoral shift of village population employment. This study proves the existence of this mediating path, which can also explain how the distance to urban centers acts as a spatial barrier to affect the development of the farmland transfer, but the results are not robust when we consider the road distance to the urban centers. The city gathers more capital, population, and information, and has stronger mobility of market factors [72]. The flow of factors from urban areas to rural areas has significant spatial heterogeneity [20], and the villages in the suburbs can obtain more off-farm employment information and opportunities than the remote villages, leading to the higher opportunity costs of agricultural labor in the surrounding villages. However, this path is affected by both urban level and transportation facilities. In the higher-level urban centers, such as prefectural centers, there are higher levels of living costs and more jobs that need advanced skills and in the lower-level urban centers, such as town centers, there are less off-farm employment. Transportation strengthens the positive influence of prefectural centers, so only county centers in this study affect the rural farmland transfer through this mediating path and are not robust when using road distance to the nearest city centers.

Finally, the cost-benefit framework is applicable to explain how urban centers influence rural farmland transfer. The farmers in the villages close to the urban centers have more off-farm employment opportunities and have a stronger willingness to obtain more economic benefits from transferring out. Besides, these villages have strong accessibility to the city and low transportation costs [73], which help transferees obtain higher agricultural income through land. This makes them more appealing, attracting more transferees interested in transferring in for agricultural production, which in turn elevates the farmland transfer rates in these villages. The policy also plays a positive role in regulation. The rural revitalization policy proposes "industrial revitalization", which encourages the development of large-scale agricultural production, trains new professional farmers, and offers economic subsidies. These measures have a positive impact on rural farmland transfer. Recent research has also begun to focus on the impact of land property rights on farmland transfer. Findings suggest that land ownership confirmation, guided by

policy, has a positive influence on rural farmland transfer [74,75]. The urban influence of this factor should be paid more attention to.

## 6. Conclusions and Policy Implication

Using Tai'an as a case, this study discusses the mechanisms of the impact of different levels of urban centers on rural farmland transfer rates. The results show that: (1) The closer to the high-level urban centers, the higher the rates of farmland transfer in the village. The county centers attract farmers to off-farm employment and the prefectural centers affect the planting structure of surrounding villages. This increases the opportunity costs of agricultural labor and increases the economic value of agricultural production. As a result, the farmland transfer rate in the village increases. (2) Different levels of urban centers have different impacts. Compared with town centers, county centers have larger off-farm employment market size and higher wage levels, and compared with prefectural centers, county centers have a lower level of living costs and more jobs that need less advanced skills can attract more surplus agricultural labor, and have a more positive impact on rural farmland transfer. (3) Town centers cannot promote off-farm employment and planting structures in the surrounding villages, which means that towns need to further explore their potential for non-agriculture industry and agricultural markets.

Based on the results, this study proposes the following suggestions: First, according to the result that there is no significant impact of the distance to town centers on rural farmland transfer, it is necessary to strengthen the economic function of towns, and fully exert the driving role of off-farm industries and agricultural markets in towns. The impact of towns on the land resource reallocation process in villages is weak, and the radiating potential of towns needs to be paid more attention to. Second, more policy support and preferential subsidies should be given to the remote villages. This involves improving the level of basic infrastructure and advocating for the scaled and intensive development of agriculture. Third, the role of village collectives and agricultural production cooperatives in land integration should be actively played. Existing studies have found that land-scale utilization promoted by village collectives and agricultural production cooperatives in remote villages can achieve greater results and improve agricultural production efficiency [76].

There are three main limitations in this study: (1) Due to the constraints of cross-sectional data, it is difficult to discuss how the urban centers affect the planting structure and off-farm employment of villages in different urbanization stages, and how this change affects the development of the farmland transfer in villages. (2) This study has found that urban centers affect farmland transfer through off-farm employment and planting structure. However, data limitations make it difficult to further discuss other mediating paths, such as new types of agriculture that include sightseeing agriculture and leisure agriculture, which need less distance to the urban centers. This is also a direction for future research. (3) In China, policies affecting land ownership can potentially influence farmland transfer rates. However, due to data limitations, this study can only discuss the impact of policies on farmland transfer from the perspective of policy incentives and economic subsidies. The influence of land ownership confirmation needs further exploration.

**Author Contributions:** Conceptualization, T.L. and M.Y.; Data curation, M.Y. and T.S.; Formal analysis, M.Y.; Funding acquisition, T.L.; Methodology, T.L. and M.Y.; Project administration, T.L.; Resources, T.L.; Software, M.Y. and T.S.; Supervision, T.L.; Validation, M.Y. and T.L.; Visualization, M.Y. and T.S.; Writing—original draft, M.Y. and T.S.; Writing—review and editing, M.Y. and T.L. All authors have read and agreed to the published version of the manuscript.

**Funding:** This research was funded by the National Natural Science Foundation of China (42371201).

**Data Availability Statement:** Not applicable.

**Conflicts of Interest:** The authors declare no conflict of interest.

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
