# Peer review of "The Heterogeneous Effects of Multilevel Centers on Farmland Transfer: Evidence from Tai’an Prefecture, China"

_land, doi:10.3390/land12101858_

Round 1

Reviewer 1 Report

The article is valuable and interesting. It fits well into the scientific profile of the journal.

Substantive comments/suggestions:

1.      I propose to soften the statement contained in the last sentence of the fourth paragraph (lines 85–86), because the term "research contradictions" is exaggerated.

2.      The categorization of cities by levels should be better explained to make the issue clearer for non-Chinese readers.

3.      Line 207 – the word "other" suggests that China belongs to the group of western countries.

4.      In the "Materials and Methods" section, it should be indicated what period of land circulation was taken into account.

5.      I suggest supplementing the description of variable X (lines 254–255) with the information that the model initially took into account the Euclidean distance, but after modifying the model, the road distance was taken into account.

6.      I propose to consider the decomposition of the M variable, because it de facto includes two variables whose impact has been quantified separately.

7.      Concerning table 1:

·        Are the given minimum and maximum values empirical in each case (it seems that in some cases values are given that the variable could potentially take)?

·        Is the control variable "Policy support" a binary variable (if so, the question arises about the meaning of the mean and standard deviation)?

8.      The interpretation of the data presented in Table 2 is complicated by the use of the same names for models 1, 3 and 5.

9.      The first sentence of the fourth paragraph in section 4.2. (line 385) requires rewording because the combination of data and symbols without their explanation and interpretation does not confirm the hypotheses.

10.   In line 389 and table 3, the value of 25.68% was given incorrectly (the partial data shows that it should be 26.68%).

11.   The title of table 1 is not very appropriate (the table does not contain information about data sets).

12.   The title of Figure 5 is too contextual.

13.   In the last paragraph of section 5, the description of the limitations of the study (especially point 2) should be improved.

Editorial comments/suggestions:

1.      Not very good quality of drawings pasted as images.

2.      No references to tables 4 and 6 in the main text.

3.      It is not advisable for a given section to end with an object (figure or table) – see sections 3.1., 3.3., 4.1., 4.2., 4.3.

4.      I propose to remove the fragment "According to the above theoretical analysis, it is proved that" (line 178), as it gives the sentence a summary character, making the sentence inappropriate to the section in which it is included.

5.      Commas between symbols in a different format (line 256).

6.      Reference to the source in superscript (line 263 and table 1).

7.      Missing period at the end of the sentence (line 311).

8.      Avoid one-sentence paragraphs (line 312).

9.      Concerning table 1:

·        For the dependent variable, I suggest using singular because there is one dependent variable in the model.

·        Is the abbreviation "S.D." means standard deviation?

10.   In the last paragraph of section 4.1. the first sentence refers to Figure 4b and the second sentence refers to Figure 4a. I suggest specifying references to the drawing appropriately.

11.   Redundant dot after the title of the journal in bibliographic description no. 1 (line 559).

Author Response

All of us appreciate for the kind suggestion. Please see the attachment.

Reviewer 2 Report

This is an interesting study exploring the impact and mechanisms of distance to urban centers on village-level farmland transfer. Especially the Authors look at the impact of distance to urban centers on farmland transfer, the differences in the impact of multilevel urban centers on farmland transfer and the mechanism of how distance to urban centers affects farmland transfer. The case study of Tai’an Prefecture, China is used for the analysis. The Authors correctly assumed the hypothesis based on a classical theory that the rate of farmland transfer in the village decreases as the distance to urban centers increases, with the village located farther from urban centers exhibiting a lower farmland transfer rate. Based on the well-designed research procedure it was found that urban centers have an impact on farmland transfer in villages, but this impact is more prominent in high-level urban centers.

The suggestions for improvement consist of

Lines 513-521 - limitations in this study to be moved to the end of the Conclusion part.

Extend Discussion by i.e. https://doi.org/10.3389/fsufs.2023.1143595 or https://doi.org/10.3390/land10010058

Author Response

(The authors gave the same response as above.)

Reviewer 3 Report

This paper discusses the influence of the distance of urban center at different levels on the transfer of agricultural land, with a novel perspective and unique views. There are the following two suggestions to the authors, please consider revising them:

 First, the author mentioned in the research hypothesis that the farther away from the urban center, the lower the transfer rate of agricultural land. The authors have only provided evidence of inferred statistics (i. e., empirical analysis), but not sensory evidence. It is suggested that the author make descriptive statistics on the data and draw pictures to prove that the farther away the village is from the city, the lower the probability of agricultural land transfer.

Second, the author proposes that the radiation effect of cities is also different because of their different levels. However, the author only uses the distance between the village and the county seat and the town center when choosing the indicators. The author should understand that most of the rural labor force is transferred to provincial capital cities or developed coastal cities, and fewer counties and towns are taken as non-agricultural employment destinations. Therefore, the author should at least consider the distance between the villages and the provincial capitals or prefecture-level cities, so as to be closer to China's reality.

Author Response

All of us appreciate for the kind suggestions. Please see the attachment.
